# ^18^F–Choline PET/CT Identifies High-Grade Prostate Cancer Lesions Expressing Bone Biomarkers

**DOI:** 10.3390/jcm8101657

**Published:** 2019-10-11

**Authors:** Nicoletta Urbano, Manuel Scimeca, Antonio Crocco, Alessandro Mauriello, Elena Bonanno, Orazio Schillaci

**Affiliations:** 1Nuclear Medicine, Policlinico “Tor Vergata”, 00133 Rome, Italy; n.urbano@virgilio.it; 2Department of Biomedicine and Prevention, University of Rome “Tor Vergata”, 00133 Rome, Italy; manuel.scimeca@uniroma2.it (M.S.);; 3Fondazione Umberto Veronesi (FUV), 20122 Milano, Italy; 4Department of Experimental Medicine, University "Tor Vergata", 00133 Rome, Italy; alessandro.mauriello@uniroma2.it (A.M.); elena.bonanno@uniroma2.it (E.B.); 5Neuromed Group, ’Diagnostica Medica’ & ’Villa dei Platani’, 83100 Avellino, Italy; 6IRCCS Neuromed, 86077 Pozzilli, Italy

**Keywords:** prostate cancer, ^18^F–choline, positron emission tomography, nuclear medicine, bone biomarkers

## Abstract

The main aim of this study was to investigate the possible association between ^18^F–choline uptake and histopathological features of prostate biopsies such as the Gleason Group and the expression of both epithelial to mesenchymal transition (vimentin) and bone mineralization (bone morphogenetics protein (BMP)-2, runt-related transcription factor 2 (RUNX2), receptor activator of nuclear factor-κB ligand (RANKL), vitamin D receptor (VDR), and pentraxin 3 (PTX3) in situ biomarkers. To this end, we enrolled 79 consecutive prostate cancer patients that underwent both the ^18^F–choline PET/CT analysis and the prostate bioptic procedure. The standardized uptake value (SUV) average values were collected from ^18^F–choline PET/CT analysis whereas Gleason Group and immunostaining data were collected from paraffin-embedded sections. Histological classification showed a heterogenous population including both low/intermediate and high-grade prostate cancers. A significant increase of ^18^F–choline uptake in high-grade prostate lesions (Gleason Score ≥8) was found. Also, linear regression analysis showed a significant correlation between ^18^F–choline uptake and the number of vimentin, RANKL, VDR, or PTX3 positive prostate cancer cells. Conversely, we observed no significant association between ^18^F–choline uptake and the expression of bone biomarkers involved in the early phases of osteoblast differentiation (BMP-2, RUNX2). In conclusion, results here reported can lay the foundation for the use of ^18^F–choline positron emission tomography (PET)/computed tomography (CT) as a diagnostic tool capable of identifying high-grade prostate cancer lesions expressing bone biomarkers.

## 1. Introduction

Cancers of the prostate represent the second cause of cancer-related death worldwide in men, with 1.6 million new cases and 366,000 deaths annually [1,2,3]. Currently, no screening programs showed benefits in terms of survival, thus the discovery of new biomarkers, as well as the development of in vivo analysis capable of identifying the prostate lesions with high metastatic potential, could represent an extraordinary possibility to reduce prostate cancer mortality. Currently, ^18^F–choline positron emission tomography (PET) is one of the most widely used diagnostic in vivo techniques for the stratifying and staging of patients affected by prostate cancer; namely in the evaluation of the disease’s extent, such as localized disease, lymph node involvement, and distance metastasis [4]. Also, ^18^F–choline can play an essential role in the identification of biopsies’ sites by detecting the regions with a higher biomarker uptake, representing higher malignant probability [5]. A disadvantage of ^18^F–choline analysis seems to be that its uptake is diagnostically relevant only after a diagnosis of biochemical relapse has been made. No less important, considering the current knowledge, ^18^F–choline PET/computed tomography (CT) is not capable of identifying recurrent disease before a rise in prostate-specific antigen (PSA) is detected [6]. In line with this, in our experience ^18^F–choline PET/computed tomography (CT) scan shows a conspicuous number of pitfalls that are easily recognizable and attributable to inflammation [7]. Given the large use of ^18^F–choline PET/CT investigations worldwide, an innovative interpretation of the choline uptake by prostate cancer cells can open new and interesting perspectives in the management of cancer patients. However, to achieve this goal, extensive correlation studies between ^18^F–choline PET/CT analysis and histopathological data are needed [8]. In this scenario, we recently reported very preliminary data about the possible use of ^18^F–choline PET/CT analysis in the detection of prostate lesions with a high propensity to develop osteoblastic bone metastatic lesions [9]. In this study, for the first time, we describe the presence of prostate cancer cells with morphological and molecular characteristics of osteoblasts (prostate osteoblast-like cells—POLCs), demonstrating the putative association between their presence and the development of bone metastasis within 5 years from histopathological diagnosis [9]. Also, we found that POLCs’ origin was linked to epithelial to mesenchymal transformation (EMT) and subsequent osteoblastic differentiation stimulated by bone morphogenetics protein (BMP)-2 [10]. 

Starting from these pieces of evidence, this study aimed to investigate the possible association between ^18^F–choline uptake, histopathological features of prostate biopsies, such as Gleason score (GS), and the expression of both EMT and bone in situ biomarkers. 

## 2. Methods

### 2.1. Prostate Samples Collection

The study was approved by the Institutional Ethical Committee of the “Policlinico Tor Vergata” (reference number # 129.18). Experimental procedures were performed in agreement with The Code of Ethics of the World Medical Association (Declaration of Helsinki). All patients have signed an informed consent prior to surgical procedures.

Only patients with a Gleason Group of at least six and/or a prostate-specific antigen (PSA) level greater than 4 ng/mL were included in the study. The exclusion criteria were a second cancer and neoadjuvant hormonal or radiation therapy prior to surgery. All patients underwent ^18^F–choline PET/CT analysis in the period from 15 to 30 days before MRI-guided biopsies. Free PSA serum concentration was determined on the day of ^18^F–choline PET/CT analysis (mean, 12.27 ng/mL ± 1.006; range, 4.23–36.30 ng/mL). 

According to inclusion and exclusion criteria, we retrospectively enrolled 79 consecutive prostate cancer patients (76.06 ± 0.93 years; range 64–82 years) underwent both ^18^F–choline PET/CT analysis and prostate bioptic procedure.

From each bioptic sample, paraffin serial sections were used for both histological and immunohistochemical investigation. 

### 2.2. ^18^F–Choline PET/CT Analysis

All patients enrolled in this study were subjected to ^18^F–methylcholine (^18^F–choline) PET/CT analysis as previously described [11]. The standardized uptake value (SUV) of the target lesion(s) was measured. Specifically, we evaluated ^18^F–choline uptake in the prostate semiquantitatively using SUVmax and SUVaverage (applying volumes of interest (VOI) with a threshold of 50%) derived from attenuation-corrected PET emission data. To reduce the operator-dependent variables, only the values of SUVmean were reported in this study. However, similar results were obtained by using SUVmax values (data not shown). Results of ^18^F–choline PET/CT (SUVaverage) were collected to verify a possible correlation between ^18^F–choline uptake in prostate tumours and both Gleason score and expression of in situ biomarkers, vimentin, BMP-2, Runt-related transcription factor 2 (RUNX2), receptor activator of nuclear factor-κB ligand (RANKL), Vitamin D Receptor (VDR), and PTX3. From each patient, the standardized uptake value (SUV) average was recorded.

### 2.3. MRI-Guided Biopsies

All patients underwent 1.5- or 3-T MRI before prostate biopsy with or without an endorectal coil. Suspicious lesions at MRI were submitted to a targeted biopsy with the use of real-time TRUS guidance using a software registration system. At least two cores were taken for each suspicious/target lesion. A correspondence between MRI target regions and uptake of choline was observed. All patients underwent a concomitant systematic biopsy at the time of the targeted biopsy, with at least six random cores taken outside the targeted biopsy area.

### 2.4. Histology

Fixation, paraffin-embedding, and hematoxylin and eosin staining were performed as previously described [12]. 

### 2.5. Immunohistochemistry

To study the immunophenotypical profile of cancer cells, we performed immunohistochemical reactions to investigate the expression of the following biomarkers: vimentin (EMT), BMP-2, RUNX2, RANKL, VDR, and PTX3 (bone metabolism). For Antigen retrieval 3 μm-thick paraffin sections were treated with Citrate pH 6.0 or EDTA citrate pH 7.8 buffers (95 °C for 30 min). Then, primary antibodies listed in Table 1 were incubated for 1 hour at room temperature. An HRP-DAB Detection Kit (UCS Diagnostic, Rome, Italy) was used to reveal the reaction of primary antibodies with their specific target. An immunohistochemical signal was assessed independently by two investigators by counting the number of positive cancer cells (out of a total of 500 in randomly selected cancer regions).

### 2.6. Statistical Analysis

We performed group-wise comparisons of the expression of the analyzed biomarkers through nonparametric Kruskal–Wallis (KW) (*p* < 0.05). Post-hoc testing was performed by Mann–Whitney test. Linear regression analyses were performed to assess the correlation between ^18^F–choline SUV average and the expression of BMP-2, RUNX2, RANKL, VDR, and PTX3 in prostate cancer tissues.

## 3. Results

### 3.1. Histological Classification

All prostate biopsies were classified in acinar adenocarcinomas according to WHO 2016 [13]. Each lesion was evaluated according to the Gleason Group (GG) classification [13]. For each patient, the highest value of GG found in biopsies of target regions has been used. In regards to the GG, we observed 13 patients with GG = 3 + 3, 16 patients with GG = 4 + 3, 26 patients with GG = 4 + 3, 15 patients with GG = 4 + 4, and 9 patients with GG = 5 + 4. In order to evaluate the association between GG and a) age; b) ^18^F–choline uptake (SUV average); and c) in situ expression of vimentin, BMP-2, RUNX2, RANKL, VDR, and PTX3, we subdivided the patients into the following groups: G_1 (patients with GG 3 + 3 or 3 + 4), G_2 (patients with GG 3 + 4), and G_3 (patients with Gleason score 4 + 4 or 5 + 4). In regards to the comparison between GG and PSA, no significant differences were observed. The baseline characteristics of patients were reported in Table 2.

### 3.2. Comparison between ^18^F–Choline Uptake and Gleason Score

In order to verify the possible predictive value of ^18^F–choline PET/CT on the prostate tumor aggressiveness and differentiation (GG value), we compared the ^18^F–choline uptake (SUV average) among the groups described above (G_1, G_2, and G_3) (Figure 1A). Specifically, a significant group effect was found (*p* = 0.0015) (Figure 1A). In regards to the post-hoc test, we observed a significant increase of SUV average in G_3 (4.96 ± 0.73) when compared to both G_2 (2.34 ± 0.35) and G_1 groups (2.94 ± 0.17) (G_1 v.s. G_3, *p* = 0.0020; G_2 v.s. G_3, *p* = 0.0014) (Figure 1B–G). Frequently, we observed patients with an SUV average of <2 in the G_1 group (Figure 1B–D) or patients with an SUV average of ≥3 in G_3 (Figure 1E–G). No others differences were found.

### 3.3. Linear Regression Analysis between ^18^F–Choline Uptake and in Situ Biomarkers

The linear regression analysis has been performed to investigate the possible association between ^18^F–choline uptake (SUV average) (Figure 2A) and the expression of both EMT and bone in situ biomarkers.

In regard to the marker of the EMT, linear regression, analyses showed a positive and significant correlation between the number of vimentin-positive prostate cancer cells and ^18^F–choline uptake evaluated in terms of SUV average (*r^2^* = 0.2787; *p* < 0.0001) (Figure 2B). Conversely, no significant difference was observed analyzing the correlation between ^18^F–choline uptake and the number of BMP-2 positive prostate cancer cells (*r^2^* = 0.01715; *p* = 0.2595) (Figure 2C). Similarly, no significant association was observed considering the expression of RUNX2 (*r^2^* = 0.03013; *p* = 0.1337) (Figure 2D). In regard to the correlation between RANKL and ^18^F–choline uptake linear regression, analysis showed a significant positive association (*r^2^* = 0.1259; *p* = 0.0017) (Figure 2E). Also, significant positive correlations were observed for both VDR (Figure 2F) and PTX3 (Figure 2G) expression and ^18^F–choline uptake. Specifically, we found the following values: VDR *r^2^* = 0.1227; *p* = 0.0019; PTX3 *r^2^* = 0.3081; *p* < 0.0001.

### 3.4. Comparison between in Situ Biomarkers and Gleason Score

We evaluated the possible association between GG and the expression of both EMT (vimentin) and mineralization (BMP-2, RUNX2, RANKL, VDR, and PTX3) biomarkers. In particular, we observed a significant group-effect in the expression of vimentin among the experimental groups (*p* < 0.0001) (Figure 3A). Expression of vimentin was higher in G_3 (297.6 ± 38.86) and G_2 (191.2 ± 16.55) as compared to G_1 (57.55 ± 10.86) groups (G_1 v.s. G_3, *p* < 0.0001; G_1 v.s. G_2, *p* < 0.0331) (Figure 3A). Also, a significant difference was observed by comparing the number of vimentin-positive cancer cells between the G_2 and G_3 groups (*p* = 0.0040) (Figure 3A). In regards to the expression of BMP-2, we found a significant group effect (*p* < 0.0001) (Figure 3B). Afterwards, in a post-hoc test, we observed a significant increase of BMP-2 positive cancer cells in both the G_2 (325.9 ± 16.71) and G_3 (332.6 ± 23.69) groups when compared to the G_1 group (93.42 ± 16.95) (G_1 v.s. G_3, *p* < 0.0001; G_2 v.s. G_1, *p* < 0.0001) (Figure 3B). A similar trend was found for the expression of RUNX2 (Figure 3C). Indeed, we observed the following significant differences: group effect *p* < 0.0001; G_1 v.s. G_3, *p* < 0.0001; G_2 v.s. G_1, *p* < 0.0001 (G_1 48.15 ± 9.82; G_2 304.3 ± 23.73; G_3 338.2 ± 31.84) (Figure 3C). The immunohistochemical analysis of RANKL showed a significant increase in the number of positive cells in groups with higher value of GG (Figure 3D). Specifically, after detection of the significant group effect (*p* < 0.0001), we observed a significant difference among the following matches: G_1 v.s. G_3, *p* < 0.0001; G_2 v.s. G_1, *p* < 0.0001; G_2 v.s. G_3, *p* = 0.0149 (G_1 77.52 ± 28.18; G_2 271.3 ± 16.20; G_3 373.0 ± 42.57) (Figure 3D). In regards to the evaluation of VDR, it is important to note that lesions in the G_1 group showed a very low number of nuclear VDR positive prostate cancer cells (Figure 3E). In detail, the statistical analysis of immunohistochemical data displayed a significative group effect (*p* < 0.0001) and significant differences for the following matches: G_1 v.s. G_3, *p* < 0.0001; G_2 v.s. G_1, *p* < 0.0001; G_2 v.s. G_3, *p* = 0.0006 (G_1 9.33 ± 3.48; G_2 213.7 ± 18.12; G_3 376.4 ± 26.54) (Figure 3E). Lastly, ANOVA analysis showed a positive group effect for the distribution of PTX3 expression in experimental group (*p* < 0.0001) (Figure 3F). Moreover, we observed a significant increase in the number of PTX3 positive cancer cells in lesions characterized by high value of GG (Figure 3F). Specifically, we noted a significant increase of PTX3 expression in G_3 (334.7 ± 39.54) when compared to both G_2 (216.0 ± 20.59) and G_1 (69.75 ± 10.93) groups (G_1 v.s. G_3, *p* < 0.0001; G_2 v.s. G_3, *p* = 0.0294) (Figure 3F). Still, a significant difference was observed between G_2 and G_1 (*p* < 0.0001) (Figure 3F).

## 4. Discussion

Recent studies emphasized the needed for innovative multidisciplinary approaches to address the new challenges of cancer medicine [15,16,17]. In particular, great relevance has been given to the collaboration between imaging diagnostics and anatomic pathology departments [17,18,19,20,21]. The comparison and sharing of data derived by imaging procedures and histopathological analysis can shed new light on cancer mechanisms, thus providing useful information for the development and/or revaluation of diagnostic analyses. Recently, we reported preliminary data about the predictive value of ^18^F–choline PET/CT analysis in the discrimination of prostate lesions with high propensity to form bone metastasis within 5 years from the pathological diagnosis [9]. From a histological point of view, we also found that prostate lesions with high propensity to form bone metastasis were characterized by the presence of cancer cells expressing EMT and bone biomarkers—the POLCs [9,10].

Given these premises, this preliminary study aimed to investigate the possible association between ^18^F–choline uptake and histopathological features of prostate biopsies such as GG and the expression of both EMT and bone in situ biomarkers. To this end, we retrospectively enrolled 79 consecutive prostate cancer patients underwent both ^18^F–choline PET/CT analysis and prostate bioptic procedure. Histological classification based on GG score showed a heterogenous population including both low/intermediate grade (G_1 and G_2) and high grade (G_3) prostate cancers. The capability of choline PET/CT analysis (both ^18^F– and ^11^C–choline) to predict histopathological features, such as GG, is a very controversial scientific issue [22,23,24]. In the study by Cimitan et al. [25], a significant correlation between the detection rate of ^18^F–choline PET/CT and the GS was found, particularly in patients with low PSA levels. Conversely, Giovacchini et al. [26] did not find any correlation between ^11^C–choline positivity and Gleason score, at multivariate analysis. Other studies [27,28] confirmed the high value of the GG for predicting the clinical outcomes of prostate cancer patients after no treatment, treatment with radical prostatectomy, or treatment with radiation therapy. Also, the GS was an independent predictor of biochemical failure for patients receiving neoadjuvant or adjuvant hormonal therapy [28]. Castellucci et al. [29] explored the role of ^11^C–choline PET/CT in detecting recurrent disease in 102 patients with a lower trigger PSA level (1.5 ng/mL). They found that only the PSA doubling time and lymph node status, not the initial GS, were significant and independent predictors of positive scan results; however, most of their patients (91/102, or 89%) had a GS of less than or equal to 7. Similarly, for 358 patients who had biochemical evidence of recurrence of prostate cancer and underwent ^11^C–choline PET/CT, authors [26] found that the GS was a less robust predictor of positive scan results than the trigger PSA; however, most of the patients (257/358, or 72%) in this series also had a GS of less than or equal to 7.

Our data displayed a significant increase in ^18^F–choline uptake only in prostate lesions with a high value of GG suggesting that an ^18^F–choline PET scan can discriminate high-grade prostate lesions from low-grade ones, but not lesions with intermediate GG values. However, if confirmed in a large-cohort population, this evidence can provide the scientific rationale for the use of ^18^F–choline PET/CT analysis as an early predictor of prostate cancer aggressiveness regardless of the PSA value. To further verify the capability of the ^18^F–choline PET/CT scan to identify lesions with a high propensity to form bone metastasis, we correlated the SUV average with the expression of in situ biomarkers that define the POLCs. In particular, we investigated the biomarkers involved in both EMT phenomenon, such as vimentin and bone metabolism [30]. Vimentin is an intermediate filament expressed in the cytoplasm of cells of mesenchymal origin [31]. It is considered one of the most important EMT markers since cytoplasm rearrangements and cytokeratin-vimentin replacement, and they occur in the very early stages of this biological process [31]. Several studies associated the acquiring of a cytoskeleton rich in vimentin filaments by cancer cells to both the increase in tumor aggressiveness [32] and the formation of the osteoblast-like cells [9,10,20,33]. Here, we found a positive significant association between ^18^F–choline uptake and the number of vimentin-positive prostate cancer cells. This data further corroborates the evidence that higher ^18^F–choline uptake can identify high-grade prostate cancer lesions. In fact, tumors characterized by numerous vimentin-positive cancer cells are classified as highly undifferentiated/aggressive lesions. Noteworthy, more than 90% of lesions with no/rare vimentin-positive prostate cancer cells showed very low uptake of choline. Thus, it is possible to hypothesize the existence of a threshold value of choline uptake capable of providing prognostic information to oncologists by predicting both the GG and the vimentin expression.

In regard to the study of bone in situ biomarkers, no significant associations were found between early markers of osteoblast differentiation (BMP-2 and RUNX2) and the choline uptake. BMP-2 is a powerful osteoblast mineralization factor that stimulates mesenchymal stem cells and pre-osteoblasts to differentiate into mature/active osteoblasts [34]. Also, its expression has been related to ectopic mineralization in breast and prostate cancers [9,10,20,33,35]. RUNX2 is a transcription factor strictly associated with osteoblast and chondrocyte differentiation [36]. Specifically, nuclear RUNX2 positivity is upregulated in pre-osteoblasts, reaches the maximal level in immature osteoblasts, and is down-regulated in mature osteoblasts [36]. In light of this, the absence of correlation between these biomarkers and choline uptake is not surprising. Indeed, it is difficult to catch the early phases of a biological phenomenon, such as POLCs formation, by using non-specific in vivo investigation. Conversely, the use of specific ligands, e.g., anti-POLCs radiolabeled antibodies, could detect prostate cancer lesions earlier than ^18^F–choline PET/CT.

^18^F–choline uptake has instead shown a positive significant association with the expression of all in situ biomarkers of mature osteoblasts (RANKL, VDR, and PTX3) that in our model underline the molecular profile of POLCs [9]. In fact, these molecules are expressed by osteoblasts during their physiological activities, i.e., bone matrix formation [37,38,39]. From the biological point of view, the uptake of ^18^F–choline in lesions characterized by a high number of POLCs (RANKL, VDR, and PTX3 expression) can be explained by the uptake of choline in cells rich in mitochondria. In the 1970s, de Ridder demonstrated that mitochondria accumulate choline against a concentration gradient in a mouse model [40]. Thus, the numerous amounts of mitochondria in osteoblasts [41] and osteoblast-like cells can explain, at least in part, the data of our study [10].

Lastly, we observed a high expression of all investigated bone biomarkers in undifferentiated prostate cancer lesions revealing the expected association between GG and immunohistochemical data.

Even though this can be considered a pilot experimental study, data reported here can provide the scientific rationale for further large-cohort population clinical studies that aim to identify SUV cut-off values for the in vivo early detection of prostate lesions with a high propensity to form bone metastasis. Indeed, as mentioned above, numerous investigations demonstrated the negative prognostic significance of the expression of bone biomarkers in human neoplasia such as prostate and breast cancers [9,10,12,17,20,30,33,35]. Specifically, the expression of these biomarkers has been associated with the development of bone metastasis within 5 years from histological diagnosis.

## 5. Conclusions

The research of new prognostic and predictive imaging biomarkers for diagnosis and prognosis of prostate cancer represents a very demanding challenge of the scientific community. However, only a few molecules investigated in preclinical animal models are successfully translated into clinical practice. Thus, an important perspective in this field could be the improvement of current knowledge, both technical and biological, on diagnostic analyses already available, such as PET/CT scan. In line with this idea, we used histopathological data to verify the possible use of ^18^F–choline PET/CT in the early identification of prostate lesions with metastatic potential evaluated in terms of GG and the presence of POLCs. Although preliminary, data reported here can lay the foundation for the use of ^18^F–choline PET/CT as a diagnostic tool capable of identifying undifferentiated prostate cancer lesions expressing bone biomarkers.

## Figures and Tables

**Figure 1 jcm-08-01657-f001:**
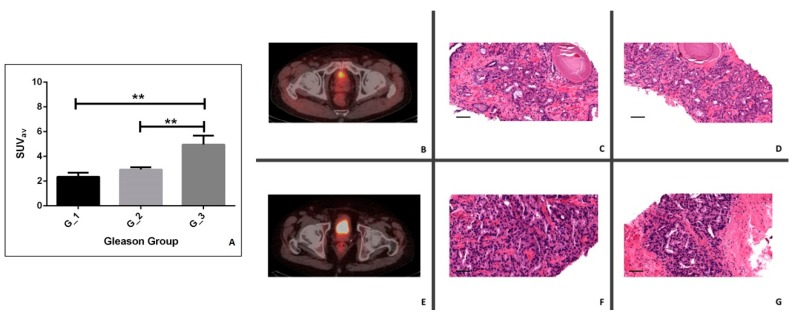
Comparison between Gleason Group (GG) and ^18^F–choline uptake. (**A**) The graph shows the standardized uptake value (SUV) average in G_1, G_2, and G_3 groups. (**B**) A 66-year-old prostate cancer patient (Gleason group of 3 + 4, primary PSA level of 7.37 ng/mL). Transaxial ^18^F–choline PET/CT image. (**C**,**D**) Prostate biopsy of lesion in panel B (Gleason Group 3 + 4). (**C**) Hematoxylin and eosin staining display a prevalent lesion with well-formed glands (Gleason score 3). (**D**) According to a Gleason score of 4, image shows poorly formed glands. (**E**) A 64-year-old prostate cancer patient (Gleason group of 4 + 4, primary PSA level of 11.25 ng/mL). Transaxial ^18^F–choline PET/CT image. (**F**,**G**) Prostate biopsy of lesion in panel E showing a 4 + 4 Gleason Group. (**F**) Hematoxylin and eosin staining display a prevalent lesion with poorly formed glands (Gleason score 4). (**G**) Image shows poorly formed glands (Gleason score 4). Scale bar represents 50 µm in each image. * *p* < 0.05, ***p* < 0.01, ****p* < 0.001.

**Figure 2 jcm-08-01657-f002:**
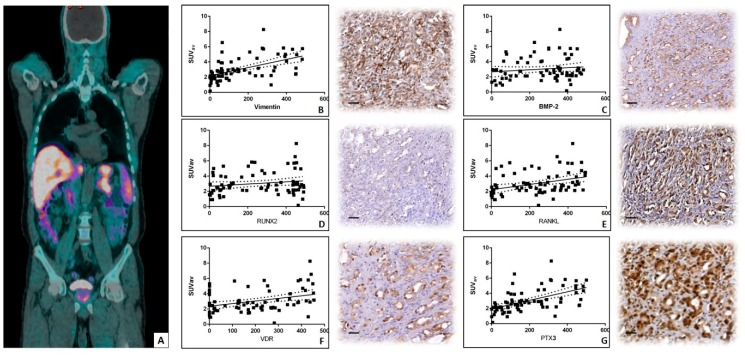
Comparison between ^18^F–choline uptake and the expression of bone biomarkers. (**A**) ^18^F-FCH PET maximum-intensity projection in a 69-year-old prostate cancer patient. (**B**) Graph displays the positive association between SUV average and the number of vimentin-positive cancer cells (*r^2^* = 0.2787; *p* < 0.0001). (**C**) No association is showed between SUV average and the number of bone morphogenetics protein (BMP)-2 positive prostate cancer cells. (*r^2^* = 0.01715; *p* = 0.2595). (**D**) No association is displayed between SUV average and the number of runt-related transcription factor 2 (RUNX2) positive cancer cells (*r^2^* = 0.03013; *p* = 0.1337). The RUNX2 reaction shows intense nuclear staining. (**E**) The graph displays the positive association between SUV average and the number of receptor activator of nuclear factor-κB ligand (RANKL) positive cancer cells (*r^2^* = 0.1259; *p* = 0.0017). (**F**) The graph shows the positive association between SUV average and the number of vitamin D receptor (VDR) positive cancer cells (*r^2^* = 0.1227; *p* = 0.0019); Intense nuclear VDR positivity is shown. (**G**) A positive association between SUV average and the number of PTX3 positive cancer cells is displayed (*r^2^* = 0.3081; *p* < 0.0001). Scale bar represents 10 µm in each image.

**Figure 3 jcm-08-01657-f003:**
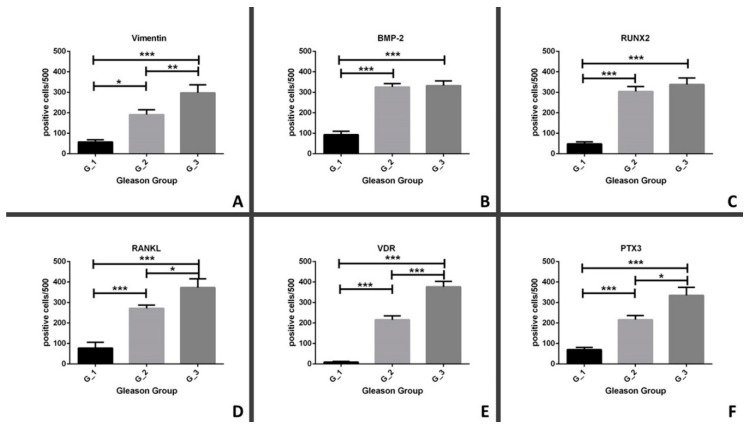
Comparison between Gleason Group and the expression of bone biomarkers. (**A**) The graph shows the number of vimentin-positive cancer cells in G_1, G_2, and G_3. (**B**) The graph displays the number of BMP-2 positive cancer cells in G_1, G_2, and G_3. (**C**) The graph shows the number of RUNX2 positive cancer cells in G_1, G_2, and G_3. (**D**) The graph displays the number of RANKL positive cancer cells in G_1, G_2, and G_3. (**E**) The graph shows the number of VDR positive cancer cells in G_1, G_2, and G_3. (**F**) The graph displays the number of PTX3 positive cancer cells in G_1, G_2, and G_3. * *p* < 0.05, ** *p* < 0.01, *** *p* < 0.001.

**Table 1 jcm-08-01657-t001:** List of primary antibodies.

Antibody	Characteristics	Dilution	Retrieval
anti-vimentin	mouse monoclonal clone V9; Ventana, Tucson, AZ, USA	Pre-diluted	EDTA citrate pH 7.8
anti-BMP2	rabbit monoclonal clone N/A; Novus Biologicals, Littleton, CO, USA	1:250	Citrate pH 6.0
anti-PTX3	rat monoclonal clone MNB1; AbCam, Cambridge, UK	1:100	Citrate pH 6.0
anti-RUNX2	Mouse monoclonal; cloneEPR14334AbCam, Cambridge, UK	1:100	Citrate pH 6.0
anti-RANKL	rabbit monoclonal clone 12A668; AbCam, Cambridge, UK	1:100	EDTA citrate pH 7.8
anti-VDR	rabbit polyclonal clone NBP1-19478; Novus Biologicals, Littleton, CO, USA	1:100	Citrate pH 6.0

**Table 2 jcm-08-01657-t002:** Baseline characteristics of patients.

	G_1 (*n* = 29)	G_2 (*n* = 26)	G_3 (*n* = 24)	*P* Value
Age	75.80 ± 2.52	76.75 ± 1.06	72.00 ± 2.58	G_1 v.s. G_2, *p* = 0.5236; G_1 v.s. G_3, *p* = 0.0971; G_2 v.s. G_3, *p* = 0.0594
PSA (ng/mL) [14]	13.23 ± 1.26	9.26 ± 2.36	13.63 ± 1.20	G_1 v.s. G_2, *p* = 0.049 *; G_1 v.s. G_3, *p* = 0.8746; G_2 v.s. G_3, *p* = 0.0551
cT/pT				
T1–T2	15 (51.7%)	13 (50%)	9 (37.5%)	/
T3–T4	13 (44.8%)	11 (42.3%)	15 (62.5%)	/
unknown	1 (3.5%)	2 (7.7%)	/	/
cN/pN				
N0	20 (68.9%)	12 (46.2%)	10 (41.6%)	/
N1	9 (31.1%)	12 (53.8%)	14 (58.4%)	/
c/M/pM				
M0	26 (89.6%)	18 (69.2%)	17 (70.8%)	/
M1	3 (11.4%)	8 (30.8%)	7 (29.2%)	/
bone lesions	2 (66.6%)	5 (62.5%)	5 (71.4%)	

* *p* < 0.05

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
