# Peer review of "18F–Choline PET/CT Identifies High-Grade Prostate Cancer Lesions Expressing Bone Biomarkers"

_jcm, 2019, doi:10.3390/jcm8101657_

Round 1

Reviewer 1 Report

Urbano et al. have described an interesting work regarding the combined used of 18F-choline PET/CT in prostate tumors to identify high risk patients to develop bone metastasis. As described throughout the manuscript, the real challenge for the urologists is to have good prognosis markers to discriminate patients that are prone to develop bone metastasis. Even though this paper is performed on a relatively small cohort for a retrospective study (79 patients) as also indicated by the authors themselves, the research is well performed and the results relatively clear cut enough to draw some pertinent conclusions.

Before giving any final decision, the authors should answer the following questions and modify some data presentation.

* line 118, it is indicated that the group G_2 corresponds to GC 3+4. It is probably a mistake. This group seems corresponding to GC 4+3. If it is the case, please correct, if not explain.

* Table 2: could it be possible to indicate the “normal values” defined in the lab for the PSA levels? Besides, are there any statistical differences in the various items in the 3 groups?

* Figures: globally the figures are too small to have a clear view (mainly the pathology figures). Histograms are as important as the scans and the tissue view. It could be proposed to re-organize these figures in order to increase the histograms/graphs size, eventually adding the examples as supplementary data. In the pathological picture, a size bar is missing.

Minor point:

* the figure legends are really repetitive and somehow boring to read.They should be rephrased.

* Typos are present within the text. E.g. “No association is showS” instead of “showED” in fig.2

Author Response

Reply to Reviewer 1

Urbano et al. have described an interesting work regarding the combined used of 18F-choline PET/CT in prostate tumors to identify high risk patients to develop bone metastasis. As described throughout the manuscript, the real challenge for the urologists is to have good prognosis markers to discriminate patients that are prone to develop bone metastasis. Even though this paper is performed on a relatively small cohort for a retrospective study (79 patients) as also indicated by the authors themselves, the research is well performed and the results relatively clear cut enough to draw some pertinent conclusions.

Reply: we would like to thank the Reviewer for expressing interest in our work, and for their availability to review our manuscript.

Before giving any final decision, the authors should answer the following questions and modify some data presentation.

* line 118, it is indicated that the group G_2 corresponds to GC 3+4. It is probably a mistake. This group seems corresponding to GC 4+3. If it is the case, please correct, if not explain.

Reply: Thanks for this point out. We are sorry for the mistake. The G_2 group corresponds to GG 3+4. In the new version of the manuscript we corrected this error.

* Table 2: could it be possible to indicate the “normal values” defined in the lab for the PSA levels? Besides, are there any statistical differences in the various items in the 3 groups?

Reply: In the new version of the manuscript we reported the reference for “normal values” defined in our lab for the PSA levels [41]. As concern age and PSA, no significant differences were found among the groups.

* Figures: globally the figures are too small to have a clear view (mainly the pathology figures). Histograms are as important as the scans and the tissue view. It could be proposed to re-organize these figures in order to increase the histograms/graphs size, eventually adding the examples as supplementary data. In the pathological picture, a size bar is missing.

Reply: Thanks for this point out. We modified the figures in order to increase the histograms/graphs size. Also, we added the scale bar for each histological image.

Minor point:

* the figure legends are really repetitive and somehow boring to read. They should be rephrased.

* Typos are present within the text. E.g. “No association is showS” instead of “showED” in fig.2

Reply: Thanks for this point out. We modified the text according to the reviewer suggestions.

Reviewer 2 Report

This study correlated SUV values of Choline-PET-CTs to Gleason score and potential prognostic biomarkers. To this end, PET-CTs were performed in prostate cancer patients in the primary situation, 2 weeks after biopsy.

The search for good and reproducible prognostic and predictive biomarkers in prostate cancer is difficult and ongoing. In this study, the SUV signal of Choline-PET was correlated to possible prognostic biomarkers predicting bone metastases, after a previous retrospective study of this group showed an association between the presence of these biomarkers in prostate biopsies and the development of bone metastases within 5 years.

Although the topic is interesting and timely, it remains confusing what the actual aim of the study is. Comparing experimental biomarkers to known clinical parameters might be a way to skip preclinical studies. However, in that case the authors should choose clinical factors that are well known to be correlated to prognosis, and not compare it to something else that is  experimental as well (average SUV). 

There are actually 2 separate things that are investigated here.

The correlation of average SUV values to Gleason score. This has been investigated several times before. However, in these studies, several factors (max SUV, mean SUV, SUV dynamics) were analyzed and compared to the gold-standard MRI and prostate specimens after radical prostatectomy. These studies showed contradicting results, resulting in the fact that that so far Choline-PET is currently not used in clinical practice to differentiate tumor aggressiveness. Please mention why you choose the mean SUV and whether you looked at other factors as well. Were the biopsies performed randomly or MRI-targeted? Since prostate cancer is mostly multifocal and the Gleason score of random biopsies regularly deviates from the definitive prostate specimen after prostatectomy, how can you be sure that the Gleason score represents the SUV? Did you have the possibility to look at the prostate specimens as well? Furthermore, was the Choline-PET also compared to the diagnostic MRI?

The correlation of average SUV to possible bone biomarkers. The authors found a correlation between average SUV values in the prostate and the presence of bone biomarkers in prostate biopsies. They concluded that PET-CT could be used as a diagnostic tool to identify high grade prostate cancer lesions that express bone biomarkers. This is nice, however, since both factors (bone biomarkers and average SUV) remain experimental, it is unclear what exactly is the use of this. For example, it is not described what the mean SUV cut-off value should be to use in clinical practice. And if there is a cut-off value, it is unclear what that says about the risk of the development of bone metastases in the future, since we do not know that of this cohort. Since these are very preliminary results, what do the authors planned to do with their results (clinical relevance)? This should be described more clearly. It is not described how the biomarkers correlated to survival, the actual development of bone metastases and whether patients had bone metastases or not. In Table 2, it is described that several patients were M1, were these bone metastases? Was there a correlation to the results as well?

General:

In general, I would advise to correct the text by a native English speaker, since there are quite a few grammatical mistakes.

Abstract:

Please change `Gleason group` to `Gleason score`, since this is the official term.

Introduction

Screening programs actually did not show a benefit in survival and it is not advised to screen PSA. Please change this sentence.

The authors describe that the 5y survival of patients with bone mets is 30%, and that bone biomarkers could reduce mortality. However, the biomarkers they describe are prognostic, not predictive biomarkers. Moreover, patients did not have primary bone metastases in this cohort, as far as it is described. Please change this sentence.

I have trouble understanding what the clinical relevance of connecting these potential biomarkers to the SUV signal of the PET-CT is. Please explain a bit better why you used the biomarkers, why you used average SUV and why it should be correlated to PET-CT.

M&M

Please describe how SUV was calculated.

Why was the PET performed after the biopsies?

Results

In section 3.1, G_2 should be `Patients with GG 4+3`, or at least, I think.

Why was age chosen as comparison factor? This is not a known influencer of Gleason score.

The authors mention the `experimental group’, but do not describe which group this is.

Discussion

It should be better described, that Gleason score has been correlated to Choline-PET before and what the results were.

I would remove the sentence about age and prostate histopathologic classification, since this is not important.

Table 2:

Please give the p-values comparing the differences between the groups

Why are some T-values missing? If all patients had biopsies, they should be at least T1c?

Why are N and M values missing when all patients received a PET-CT?

Figure 1:

The graph is too small compared to the pictures.

Author Response

Reply to Reviewer 2

This study correlated SUV values of Choline-PET-CTs to Gleason score and potential prognostic biomarkers. To this end, PET-CTs were performed in prostate cancer patients in the primary situation, 2 weeks after biopsy.

The search for good and reproducible prognostic and predictive biomarkers in prostate cancer is difficult and ongoing. In this study, the SUV signal of Choline-PET was correlated to possible prognostic biomarkers predicting bone metastases, after a previous retrospective study of this group showed an association between the presence of these biomarkers in prostate biopsies and the development of bone metastases within 5 years.

Although the topic is interesting and timely, it remains confusing what the actual aim of the study is. Comparing experimental biomarkers to known clinical parameters might be a way to skip preclinical studies. However, in that case the authors should choose clinical factors that are well known to be correlated to prognosis, and not compare it to something else that is experimental as well (average SUV).

Reply: we would like to thank the Reviewer for his/her availability to review our manuscript. We hope that the modified version of our manuscript can meet his/her comments.

There are actually 2 separate things that are investigated here.

The correlation of average SUV values to Gleason score. This has been investigated several times before. However, in these studies, several factors (max SUV, mean SUV, SUV dynamics) were analyzed and compared to the gold-standard MRI and prostate specimens after radical prostatectomy. These studies showed contradicting results, resulting in the fact that that so far Choline-PET is currently not used in clinical practice to differentiate tumor aggressiveness. Please mention why you choose the mean SUV and whether you looked at other factors as well. Were the biopsies performed randomly or MRI-targeted? Since prostate cancer is mostly multifocal and the Gleason score of random biopsies regularly deviates from the definitive prostate specimen after prostatectomy, how can you be sure that the Gleason score represents the SUV? Did you have the possibility to look at the prostate specimens as well? Furthermore, was the Choline-PET also compared to the diagnostic MRI?

Reply:  Thanks for these comments that can improve our manuscript.

As concern the evaluation of SUV, we acquired both the max and average SUV for each patient. We obtained similar data using SUV max but, in our opinion, the value of SUVmax is most affected by operator-dependent variables than SUVaverage. We added these considerations in the main text (Material and Methods paragraph). Conversely, for  our case selection we do not have the value of dynamic SUV. 

Prostate biopsies were MRI-targeted according to the protocol of our department. Specifically, according to the MRI images region/s target/s was/were identified and biopsies were performed both at target regions and non-target regions (at least 5 for each regions). For this study, we evaluated the biopsies performed in the target regions considering the more higher Gleason Group for each patient. Also, in our case selection, target regions identify with MRI studies corresponded with the regions of choline uptake. No further associations between PET and MRI data were performed (this wasn’t our aim). We added these considerations in the main text.

The correlation of average SUV to possible bone biomarkers. The authors found a correlation between average SUV values in the prostate and the presence of bone biomarkers in prostate biopsies. They concluded that PET-CT could be used as a diagnostic tool to identify high grade prostate cancer lesions that express bone biomarkers. This is nice, however, since both factors (bone biomarkers and average SUV) remain experimental, it is unclear what exactly is the use of this. For example, it is not described what the mean SUV cut-off value should be to use in clinical practice. And if there is a cut-off value, it is unclear what that says about the risk of the development of bone metastases in the future, since we do not know that of this cohort. Since these are very preliminary results, what do the authors planned to do with their results (clinical relevance)? This should be described more clearly. It is not described how the biomarkers correlated to survival, the actual development of bone metastases and whether patients had bone metastases or not. In Table 2, it is described that several patients were M1, were these bone metastases? Was there a correlation to the results as well?

Reply:  Thanks for these comments that can improve our manuscript.

The study here proposed is a pilot investigation about the possible association between choline uptake and the expression of bone mineralization in prostate cancer. Several studies demonstrated that the expression of these biomarkers in prostate and other cancers can be related to the osteotropism of tissues (see reference 8-10, 16-20, 28, 31,32). In particular, we and our groups demonstrated the association among the expression of these biomarkers, the presence of osteoblast-like cells and the development of bone metastasis, both in prostate and breast cancer, within of 5-years from histological diagnosis. Thus, data here showed can lay the foundation for further studies, on large cohort population, that aim to identify SUV cut-off values for the identifications of prostate lesions with high propensity to form bone metastasis. The experimental nature of our study has been better elucidated in the revised form of our manuscript. Also, we added comments on the possible future clinical applications of our data. Unfortunately, we haven’t got long term follow-up for these patients, so survival data aren’t available. Lastly, we reported how much bone metastatic lesions are present in our case selection at the time of PET (see table2). However, the limited number of analyzed patients don’t allow an accurate statistical analysis of patients with bone lesion as compared patients without bone metastasis. 

General:

In general, I would advise to correct the text by a native English speaker, since there are quite a few grammatical mistakes.

Reply:  We performed a complete revision of the manuscript. 

Abstract:

Please change `Gleason group` to `Gleason score`, since this is the official term.

Reply:  done.

Introduction

Screening programs actually did not show a benefit in survival and it is not advised to screen PSA. Please change this sentence.

Reply:  thanks for this point out. We deleted this sentence.

The authors describe that the 5y survival of patients with bone mets is 30%, and that bone biomarkers could reduce mortality. However, the biomarkers they describe are prognostic, not predictive biomarkers. Moreover, patients did not have primary bone metastases in this cohort, as far as it is described. Please change this sentence.

Reply:  thanks for this point out. We deleted this sentence.

I have trouble understanding what the clinical relevance of connecting these potential biomarkers to the SUV signal of the PET-CT is. Please explain a bit better why you used the biomarkers, why you used average SUV and why it should be correlated to PET-CT.

Reply:  thanks for this point out. In the revision form of our manuscript we added comments about the clinical relevance of our study also better specifying the rationale of the study.

M&M

Please describe how SUV was calculated.

Reply: In the new version of our manuscript we added a description of the SUV calculation in the M&M section.

Why was the PET performed after the biopsies?

Reply: thanks for highlighted this error. We corrected it

Results

In section 3.1, G_2 should be `Patients with GG 4+3`, or at least, I think.

Reply: thanks for highlighted this error. We corrected it

Why was age chosen as comparison factor? This is not a known influencer of Gleason score.

Reply:  thanks for this point out. We deleted the data relative to comparison between age and GS.

The authors mention the `experimental group’, but do not describe which group this is.

Reply: thanks for highlighted this error. We corrected it

Discussion

It should be better described, that Gleason score has been correlated to Choline-PET before and what the results were.

Reply:  thanks for this point out. In the revision form of our manuscript we added comments about the previously results of the correlation between Gleason score and Choline-PET.

I would remove the sentence about age and prostate histopathologic classification, since this is not important.

Reply:  done.

Table 2:

Please give the p-values comparing the differences between the groups

Reply:  we reported the p values for age and PSA.

Why are some T-values missing? If all patients had biopsies, they should be at least T1c?

Reply: Unfortunately, in histological report of patients enrolled retrospectively same information was missing.

Why are N and M values missing when all patients received a PET-CT?

Reply: thanks for highlighted the errors relative to the N values (we had based of analysis on histological report rather than PET/CT); Conversely, Mx was already zero in each group. We corrected the table 2.

Figure 1:

The graph is too small compared to the pictures.

Reply: we have enlarged the graph.

Round 2

Reviewer 2 Report

The authors now improved their paper significantly, all comments are answered sufficiently.